# Elaborate cellulosome architecture of *Acetivibrio cellulolyticus* revealed by selective screening of cohesin–dockerin interactions

Yuval Hamberg[1], Vered Ruimy-Israeli[1], Bareket Dassa[1], Yoav Barak[1,2], Raphael Lamed[3], Kate Cameron[4], Carlos M.G.A. Fontes[4], Edward A. Bayer[1] and Daniel B. Fried[1]

[1] Department of Biological Chemistry, The Weizmann Institute of Science, Rehovot, Israel
[2] Chemical Research Support, The Weizmann Institute of Science, Rehovot, Israel
[3] Department of Molecular Microbiology and Biotechnology, Tel Aviv University, Ramat Aviv, Israel
[4] CIISA – Faculdade de Medicina Veterinária, Universidade de Lisboa, Avenida da Universidade Técnica, Lisboa, Portugal

Corresponding authors
Edward A. Bayer,
ed.bayer@weizmann.ac.il
Daniel B. Fried,
danielbennett.fried@gmail.com

## ABSTRACT

Cellulosic waste represents a significant and underutilized carbon source for the biofuel industry. Owing to the recalcitrance of crystalline cellulose to enzymatic degradation, it is necessary to design economical methods of liberating the fermentable sugars required for bioethanol production. One route towards unlocking the potential of cellulosic waste lies in a highly complex class of molecular machines, the cellulosomes. Secreted mainly by anaerobic bacteria, cellulosomes are structurally diverse, cell surface-bound protein assemblies that can contain dozens of catalytic components. The key feature of the cellulosome is its modularity, facilitated by the ultra-high affinity cohesin–dockerin interaction. Due to the enormous number of cohesin and dockerin modules found in a typical cellulolytic organism, a major bottleneck in understanding the biology of cellulosomics is the purification of each cohesin- and dockerin-containing component, prior to analyses of their interaction. As opposed to previous approaches, the present study utilized proteins contained in unpurified whole-cell extracts. This strategy was made possible due to an experimental design that allowed for the relevant proteins to be "purified" via targeted affinity interactions as a function of the binding assay. The approach thus represents a new strategy, appropriate for future medium- to high-throughput screening of whole genomes, to determine the interactions between cohesins and dockerins. We have selected the cellulosome of *Acetivibrio cellulolyticus* for this work due to its exceptionally complex cellulosome systems and intriguing diversity of its cellulosomal modular components. Containing 41 cohesins and 143 dockerins, *A. cellulolyticus* has one of the largest number of potential cohesin–dockerin interactions of any organism, and contains unusual and novel cellulosomal features. We have surveyed a representative library of cohesin and dockerin modules spanning the cellulosome's total cohesin and dockerin sequence diversity, emphasizing the testing of unusual and previously-unknown protein modules. The screen revealed several novel cell-bound cellulosome architectures, thus expanding on those previously known, as well as

soluble cellulose systems that are not bound to the bacterial cell surface. This study sets the stage for screening the entire complement of cellulosomal components from *A. cellulolyticus* and other organisms with large cellulosome systems. The knowledge gained by such efforts brings us closer to understanding the exceptional catalytic abilities of cellulosomes and will allow the use of novel cellulosomal components in artificial assemblies and in enzyme cocktails for sustainable energy-related research programs.

## INTRODUCTION

The anaerobic cellulolytic bacterium *A. cellulolyticus* expresses a cell–surface bound multi-enzyme complex known as the cellulosome—a highly elaborate nanomachine that efficiently degrades crystalline cellulose (*Bayer, Kenig & Lamed, 1983*; *Lamed, Setter & Bayer, 1983*; *Shoham, Lamed & Bayer, 1999*). The proteins that make up the cellulosome consist of enzymes and non-catalytic scaffoldins that each contains at least one dockerin and/or one cohesin module, respectively (*Bayer et al., 2004*). The ultra-high affinity cohesin–dockerin interaction between cellulosomal components allows for the assembly of complex branching cellulosome architectures (*Fierobe et al., 1999*). Cellulosome assembly is dictated by the specific interactions between cohesins and dockerins with different affinity profiles (*Haimovitz et al., 2008*; *Noach et al., 2003*). In *A. cellulolyticus*, these interactions can generally be classified as either type I or type II, based on sequence homology alignments (*Dassa et al., 2012*), but experimental examination of the variety of cohesin–dockerin interactions is necessary to define their full affinity profiles. Knowledge of these specificities is needed in order to predict how the scaffoldin (Sca) proteins connect with one another to form cellulosome architectures.

Studies completed before sequencing of *A. cellulolyticus* genome revealed that the bacterium presents at least two different cellulosomes on its surface (Fig. 1) (*Xu et al., 2004*). Cellulosome A is anchored to the cell surface through the surface-layer homology (SLH) domains (*Chauvaux, Matuschek & Beguin, 1999*; *Lemaire et al., 1995*; *Zhao et al., 2006*) of ScaC, and cellulosome B is anchored to the cell surface through the SLH of ScaD (*Pinheiro et al., 2009*). Since the anchoring scaffoldins of *A. cellulolyticus* have different structures and cohesin–dockerin interaction profiles, they create cellulosomes with different architectures. Cellulosome A contains three kinds of scaffoldin proteins, including the singular ScaB, which acts as an adaptor scaffoldin between the type-I cohesins of ScaC and the type-II dockerin of ScaA (*Ding et al., 1999*; *Xu et al., 2003*). Cellulosome B contains two kinds of scaffoldin proteins, the surface-bound ScaD, which contains both type-I and type-II cohesins, and ScaA (*Xu et al., 2004*). Both cellulosomes

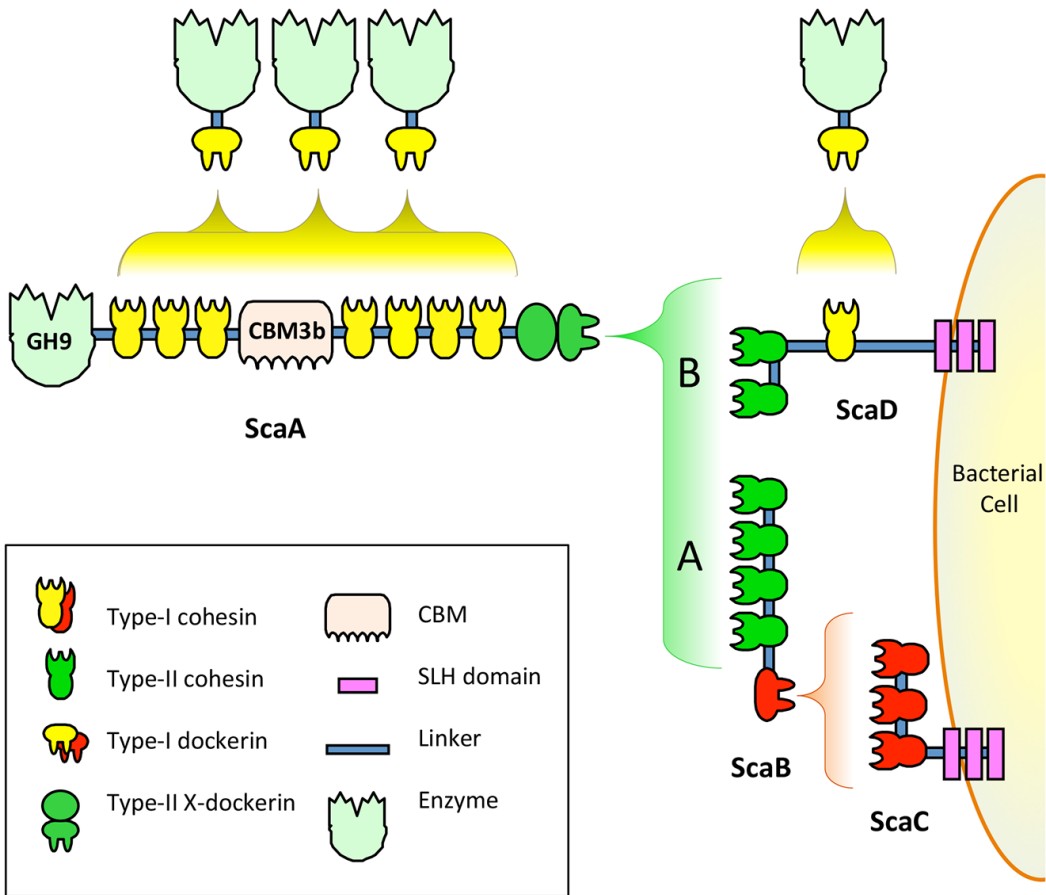

**Figure 1** **Model of previously studied *A. cellulolyticus* cellulosome architectures. Cellulosome A** comprises ScaA, ScaB, and ScaC, which bind one another through specific cohesin–dockerin interactions. ScaA contains a type I cohesin array which allows seven dockerin-containing enzymes to dock onto the cellulosome. ScaA also contains a cellulose-binding module (CBM) and a family 9 glycoside hydrolase catalytic module (GH9). ScaB acts as an adaptor between ScaA and ScaC, and allows four ScaA proteins to bind to its type II cohesins. Another type I scaffoldin, ScaC, can selectively bind three ScaB proteins and attaches the entire cellulosome to the cell surface via its surface-layer homology (SLH) domains. **Cellulosome B** contains two scaffoldins, ScaA and ScaD. ScaD also anchors the cellulosome to the cell surface via surface-layer homology (SLH) domains and can bind two ScaA proteins by its two type II cohesins. It can also bind a type I dockerin-containing enzyme via its type I cohesin. (Figure adaptation from *Xu et al., 2004*).

A and B include multiple copies of ScaA, which is decorated with a family 9 glycoside hydrolase (GH9) catalytic module, nine cohesins and a cellulose-binding module (CBM).

Recent genome-wide sequencing (*Hemme et al., 2010*) and bioinformatics analysis (*Dassa et al., 2012*) of *A. cellulolyticus* have revealed that *A. cellulolyticus* produces a much more elaborate cellulosome system than previously considered, comprising 143 dockerin containing genes and 41 cohesins, found on 16 scaffoldin proteins (Fig. 2A). In addition to its large size and complexity, some striking features of the *A. cellulolyticus* cellulosome are a number of X-dockerin modular dyads, proteins with unusual cohesin and dockerin

## A. Scaffoldins and cohesin-containing proteins

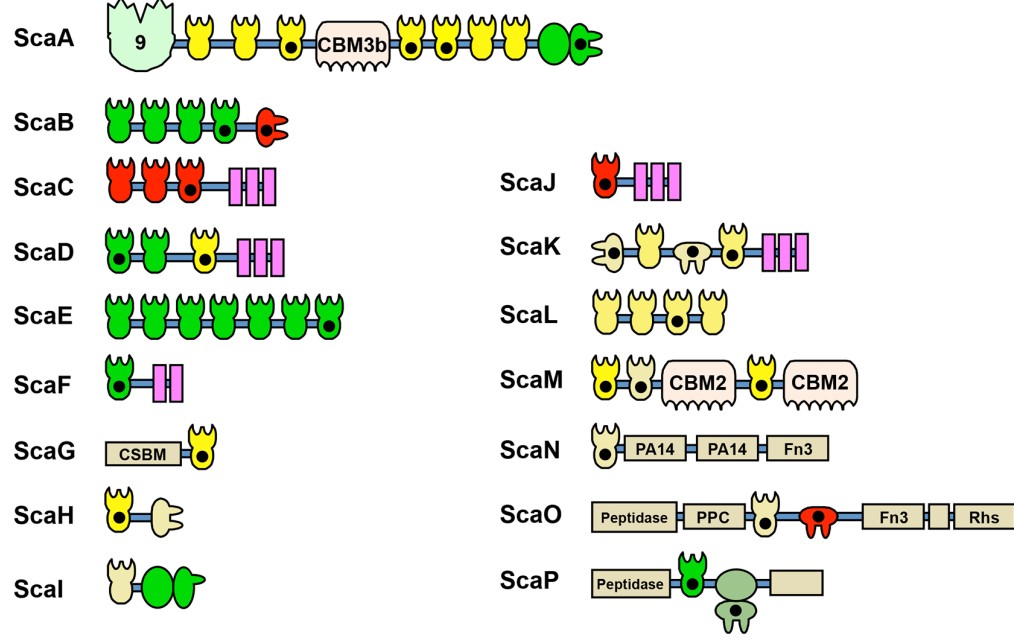

## B. Dockerin-containing proteins and enzymes

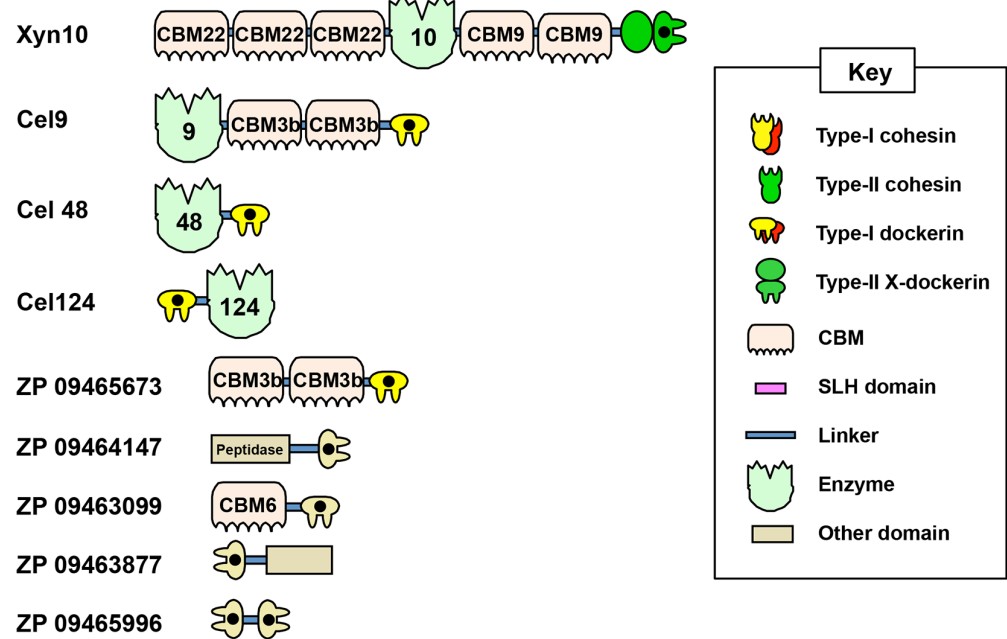

**Figure 2 Modular architecture of *A. cellulolyticus* scaffoldins and cohesin-containing proteins (A), and dockerin-containing proteins and enzymes selected for the screen (B).** Protein module symbols are indicated in the key and are color coded based on measured affinity profile as described in Table 1. Modules studied in this work are indicated with a black dot (*Dassa et al., 2012*).

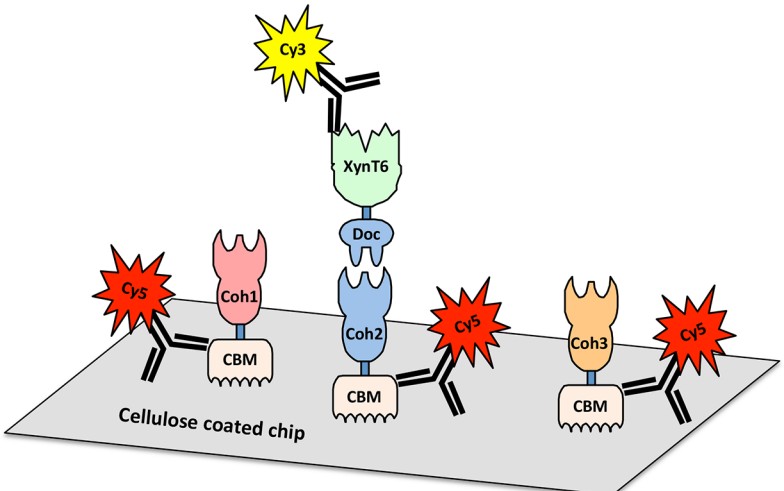

**Figure 3 Scheme for cellulose-chip-based fluorescent antibody probing of cohesin–dockerin interactions.** Cohesins of interest (Coh1, Coh2, and Coh3) fused to cellulose-binding modules (CBMs) are bound to the cellulose-coated chip. A dockerin of interest (Doc), fused to the xylanase XynT6, interacts only with specific cohesins (Coh2, in this case). Cohesin-dockerin interaction is measured by the co-localization of Cy5-conjugated anti-CBM antibodies and Cy3-conjugated anti-XynT6 antibodies. Cy5 fluorescence represents general protein attachment to the chip, whereas Cy3 fluorescence represents specific, dockerin-induced interaction with the cohesin partner.

arrangements, and cellulolytic components fused to normally non-cellulosomal proteins. The main goal of the current study was to investigate ligand specificity of cohesins from all scaffoldins of *A. cellulolyticus*, as well as a selection of dockerins from cellulosomal proteins (Fig. 2B). To investigate this architectural diversity, several specialized techniques were developed. To increase expression and solubility of recombinant cohesin and dockerin modules, a standard *Eschericia coli* expression cassette system has been created, in which cohesins and dockerins are fused to a CBM and xylanase (Xyn) module, respectively (*Barak et al., 2005*). These chimeras, designated CBM-cohesin (CBM-Coh) and xylanase-dockerin (Xyn-Doc) can be used in ELISA-based and blotting experiments (*Haimovitz et al., 2008*; *Noach et al., 2003*) in order to screen for interactions between cohesin–dockerin pairs (Fig. 3) (*Barak et al., 2005*). The approach is especially appropriate for affinity blotting, where the CBM-Coh is selectively immobilized onto a cellulose-coated chip, and after application of the Xyn-Doc, the cohesin–dockerin interaction is determined by anti-CBM and anti-Xyn fluorescence-conjugated-antibody staining. Utilizing a double staining approach allows us to determine the levels of both cohesin and dockerin in the assembled protein complexes. In addition, the present study has incorporated computational tools to create an improved screening method for broad analysis of cohesin–dockerin interactions. By identifying the connectivities between representative cohesins and dockerins from the *A. cellulolyticus* cellulosomes, this study elucidated several new cellulosome systems in *A. cellulolyticus*, and elaborated our knowledge of the previously known cellulosomes.

## MATERIALS & METHODS

### Module selection and cloning

Cellulosome-encoding regions of the *A. cellulolyticus* genome, identified as cohesin or dockerin modules, were selected in this study based on previous bioinformatics work (*Dassa et al., 2012*) (Fig. 2). In order to ensure correct expression, folding, or function of the modules, several flanking amino acids near both termini of the sequence were included.

*A. cellulolyticus* ATCC 33288 genomic DNA was generously supplied by Harish Kumar from the Lamed lab at Tel Aviv University. Cohesin and dockerin genes were amplified from the genomic DNA using PCR primers (See Tables S1A and S1B, respectively) and were subsequently inserted into a previously-constructed pET28A- and pET9-based vector system to express CBM-fused cohesins and xylanase-fused dockerins, respectively (*Barak et al., 2005*). The plasmids were double digested with *Bam*HI and *Xho*I and the corresponding double-digested gDNA PCR products were ligated into the appropriate linearized plasmid using T4 ligase (Fermentas, Vilnius, Lithuania) at 16 °C for 2 h. The ligation product was then transformed into chemically competent *E. coli* XL-1 cells and grown overnight on 50 mg/L kanamycin-containing agar media. Colonies were subsequently used to inoculated 3 ml of 50 mg/L kanamycin-containing LB media, and the plasmid was isolated using the DNA miniprep kit (Qiagen) and subjected to DNA sequencing analysis by the Weizmann Institute sequencing unit.

### Protein overexpression and lysate extraction

Plasmids containing the fusion proteins were used to transform competent BL-21 *E. coli* cells that were grown on kanamycin-containing agar plates overnight. Colonies were picked and individually inoculated into a 48-well plate (Axygen 48 deep wells plate #P-5ML-48-C). The protein autoinduction expression media consisted of 2 ml of auto-inducing ZYP-5052 rich medium with 60 μg/ml of kanamycin (*Studier, 2005*). After overnight incubation at 37 °C with shaking at 200 rpm, the 48-well plate was centrifuged at 3700 g for ten minutes, and the supernatant was discarded. The pellet was resuspended in pickup buffer (10% pop culture (Novagen, Millipore, Billerica, MA) and 90% tris-buffered saline (TBS) containing 5 μM phenylmethanesulfonylfluoride (PMSF), 2 μM benzamidine, 300 nM benzamide, and 20 μg lysozyme. The plate was incubated for 2–3 h at 16 °C and then centrifuged at 3700 g for 1 h.

### Fluorescent labeling of antibodies

Polyclonal rabbit anti-Xyn T6 antibody from *Geobacillus stearothermophilus* and polyclonal rabbit anti-CBM T6 antibody from *Clostridium thermocellum* were used as previously described (*Morag et al., 1995*). N-hydroxysuccinimide-ester-activated Cy5 dye and Cy3 dyes (GE Healthcare) were resuspended in 0.1 M sodium carbonate buffer (pH 9) and conjugated to the antibody (1 mg/ml), according to the manufacturer's instructions. Free dye was removed by dialysis against TBS. The Cy-labeled antibodies were stored in 50% glycerol at −20 °C.

## Microarray methods

Microarray preparation was accomplished using a MicroGrid II high-throughput automated microarrayer (BioRobotics Digilab, Marlborough, MA). The CBM-Coh lysate was diluted in TBS by factors of four giving dilutions of 1:16, 1:64, 1:256, 1:1,024, and 1:4,096, and was then printed in quadruplicate onto a cellulose-coated GSRC-1 glass slide (Advanced Microdevices Pvt. Ltd., Ambala, Cantt, India) using a $6 \times 4$ solid pin array with a spot diameter of 0.2 mm at 0.375 mm intervals. The slide was then incubated in blocking buffer (TBS, 1% bovine serum albumin (BSA), 10 mM $CaCl_2$, 0.05% Tween 20) at room temperature for 30 m. After blocking, the slide was incubated for ten minutes with the desired Xyn-Doc-containing lysate, and diluted by 10 000-fold in TBS. The plate was then washed three times (ten minutes each) with washing buffer (TBS with 10 mM $CaCl_2$, 0.05% Tween 20). Fluorescent staining was performed by incubating the slide for 30 m with 60 μM Cy3-labeled anti-Xyn T6 antibody and 45 μM Cy5-labeled anti-CBM antibody in blocking buffer. The slide was washed again, air-dried, and scanned for fluorescence signals using a Typhoon 9400 Variable Mode Imager I (GE Healthcare Bio-Sciences AB, Uppsala, Sweden). Scan results were saved as 16 bit gray-scale files containing two channels corresponding to the Cy3 and Cy5 fluorescent emissions.

Microarray scan results were processed using ImageJ (http://imagej.nih.gov/ij/) in order to quantify the fluorescence intensity of each protein spot. The averaged ratio of Cy3 to Cy5 fluorescence signal was then determined and graphed, with the highest measurement for each plate (excluding the XynCBM control) normalized to a value of 1.

# RESULTS AND DISCUSSION

## Module selection

Based on previous bioinformatics works, many of the cohesins and dockerins of *A. cellulolyticus* form architectures never observed before in cellulosomes. For example, ScaK contains the rare feature of having two dockerins in the same polypeptide. ScaM harbors an unusual CBM (from family 2, not previously observed in cellulosomal components), and ScaO and ScaP uniquely contain putative peptidases in their scaffoldins. Additionally, some scaffoldins contain unusual X-dockerin modular dyads (*Dassa et al., 2012*). With its exceptionally large number of cohesins and dockerins, as well as its unique architectural features, the cellulosomes of *A. cellulolyticus* are prime candidates for an organism-wide high-throughput screen of cohesin–dockerin interactions. From the total 5863 possible cohesin–dockerin combinations presented by *A. cellulolyticus*, this study selected and studied a smaller, representative protein library for the development of a medium-throughput screen. By selecting and testing the most unusual modules, as well as those from key cellulolytic enzymes, we hoped to gain further insight into the unique features of the *A. cellulolyticus* cellulosome systems. The scaffoldins and other cohesin-containing proteins of *A. cellulolyticus*, along with nine additional dockerin-containing proteins selected for this study are shown in Fig. S1.

Selection of cohesins to be included in the screen was based on the following considerations: (1) In order to represent all the scaffoldins, one or more cohesins from

each of the 16 scaffoldins were selected (with the exception of ScaI, which does not contain a cell secretion signal peptide). (2) Previously studied cohesins were selected to function as positive or negative controls. (3) Cohesins with highly divergent sequences found within the same scaffoldin were selected in order to determine any differences in binding. (4) Cohesins located near the scaffoldin termini were preferentially selected, due to difficulties associated with cloning from highly repetitive regions of the scaffoldin genes. Additionally, CohA4 and CohA5 were tested both separately and as a pair (Coh4 + 5). This was done in order to check whether the pair functioned cooperatively, due to the unusually short linker between the two cohesins. In total, 21 cohesin constructs were selected for the screen.

Dockerins were selected based on the following considerations: (1) Dockerins contained within scaffoldin proteins were selected due to their presumed importance in architectural connectivity. (2) Dockerins contained in putative enzymes were selected in order to investigate the location of catalytic subunits within cellulosomes. In this context, dockerins from the major cellulosome-associated exoglucanase (Cel48A), a family 9 endoglucanase, the recently described family 124 enzyme and an X-dockerin of a family 10 xylanase were included in this study. (3) Dockerins found in interesting non-canonical configurations were selected, such as the highly unusual double dockerin construct (DocDoc ZP 09465996). (4) Dockerins containing unusual amino acid sequences at their putative cohesin-recognition interface were selected. Such dockerins diverged from the canonical dual calcium binding and cohesin binding sequences, specified in *Dassa et al. (2012)*. Some of the latter dockerins were parts of proteins that bear unusual modular content, e.g., CBM/s exclusively or a peptidase, and in one case, the potential binding characteristics of a double dockerin was explored. In total, 15 dockerin constructs were selected for the screen.

## Prototype methodology designed for future high-throughput screening

Focusing our screen on 21 cohesins and 15 dockerins supported the analysis of a total of 315 cohesin–dockerin interactions to be tested. In order to test this large number of interactions, several procedural measures were taken to streamline the protein expression and testing steps. (1) As described in previous work from our lab, cohesins and dockerins were incorporated into standardized protein expression cassettes containing an N-terminally fused CBM, and dockerins were incorporated into standardized protein expression cassettes containing an N-terminally fused xylanase (*Barak et al., 2005*). This previously developed system allowed for a simplified and standardized cloning and expression workflow. (2) Protein expression was performed in autoinduction expression medium to increase experimental efficiency and to increase protein expression levels over IPTG-induced protein expression. (3) In a critically relevant approach, cohesin–dockerin interaction experiments were performed using crude cell lysates instead of nickel-NTA-purified protein solutions in order to eliminate purification steps. The effective use of crude cell lysates was enabled by our unique experimental design. Protein screening on a cellulose-coated chip allows the CBM-cohesin to bind to the chip surface at the beginning of the experiment, thereby presenting the cohesin module to the

medium. Washing steps act as cohesin purification steps. With the addition of the crude xylanase-dockerin, subsequently bound dockerin is again effectively purified by washing steps. The binding of the CBM to the cellulose surface indicates that its activity remains intact and gives a level of confidence that the fused cohesin module would also be active (*Berdichevsky et al., 1999*; *Ofir et al., 2005*). The grafting of the dockerin module onto the *G. stearothermophilus* xylanase T6 is also considered to stabilize the structure and activity of the dockerin (*Barak et al., 2005*; *Handelsman et al., 2004*).

To validate that lysate-derived protein could be used to differentiate between positive and negative cohesin–dockerin binding, a cross-reactivity experiment was conducted between proteins from *C. thermocellum* and *A. cellulolyticus*. For this experiment the *C. thermocellum* XynDoc contained Doc48S and the *A. cellulolyticus* XynDoc contained DocB (from ScaB). Because cohesin–dockerin specificity is sequence specific, this test could determine any binding specificity differences between lysate-derived and purified protein. Results of this experiment showed that in general cell lysates and purified protein both had the expected species-specific binding profiles. For the *A. cellulolyticus* DocB, several positive species-specific binding interactions were observed, as expected, including interactions with CohC3 and CohJ. Both lysate and purified proteins showed similar binding results. *A. cellulolyticus* DocB showed reduced binding to the *C. thermocellum* CohA2 control, as expected (Figs. 4A and 4B).

Control experiments testing the binding of *C. thermocellum* Doc48S with lysate-derived and purified proteins showed little or no cross-interactivity with *A. cellulolyticus* cohesins (Figs. 4C and 4D). Low levels of cross-species dockerin binding were observed for CohM2, CohD1, and CohL3, although this is probably indicative of nonspecific binding associated with these cohesins (see below).

To develop a potential high-throughput method for testing cohesin–dockerin interactions, a microarray spotting robot was used to immobilize overexpressed CBM-Cohs onto cellulose-coated chips. This approach is advantageous, since the CBM of the fusion protein allows selective interaction with the cellulosic surface of the chip, thus orienting the cohesin in a solution-exposed position for facile interaction with the dockerin probe. Although the CBM-Coh is part of a whole-cell lysate and "contaminated" by an abundance of other host-cell proteins, it is essentially "purified" through its selective interaction with the chip and subsequent washing steps. The chips were then interacted with different Xyn-Doc solutions. Those dockerins that interact selectively with the test cohesin will essentially be affinity-purified on the chip, and the rest of the irrelevant cell-derived proteins will be discarded in the wash steps. Cohesin-dockerin interactions were detected by co-staining the chips with Cy5-conjugated anti-CBM and Cy3-conjugated anti-xylanase antibodies, respectively. Cy5 staining indicated the general expression of the CBM-fused cohesin that was bound to the cellulose-coated slides, and the Cy3 labeling indicated the amount of xylanase-fused dockerin that bound to the respective cohesins. This double-staining approach allowed for differences in cohesin and dockerin protein levels to be normalized. The ratio of Cy5 and Cy3 staining was thus used to determine relative interaction strength between the cohesins and dockerins. The resulting measurements gave

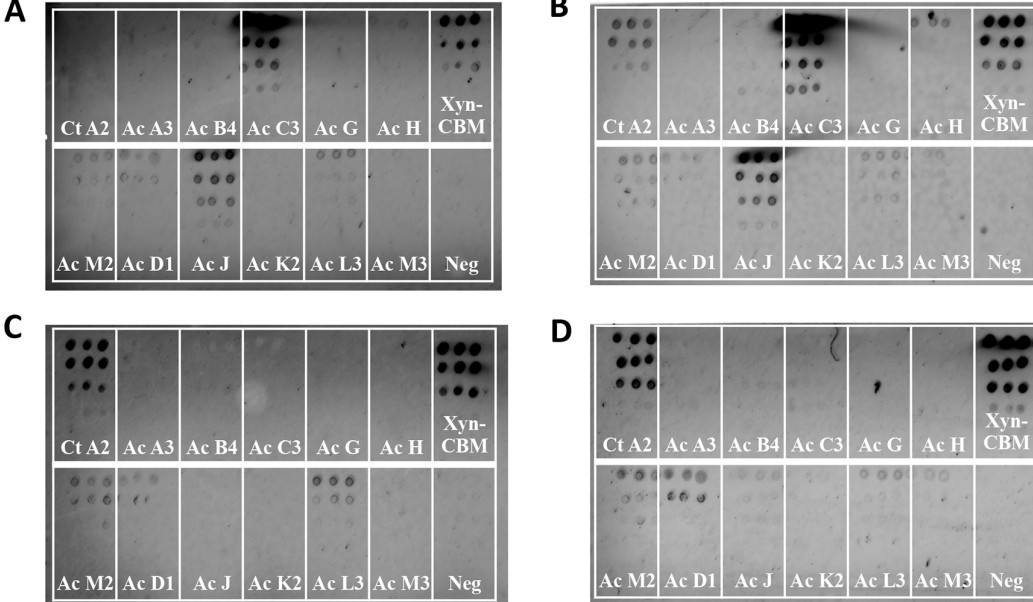

**Figure 4 Test for cross-species cohesin–dockerin interactivity using lysate-derived dockerin or purified dockerin—representative examples.** Cy3 fluorescence scan of cellulose-coated slides containing interacting cohesin–dockerin pairs. Each box contains five four-fold serial dilutions of the indicated CBM-Coh, blotted vertically in triplicate. Spotting with CBM-Coh proteins was followed by interaction with either lysate-derived Xyn-dockerin (A, C) or Ni-NTA purified Xyn-dockerin (B, D) for comparison. Cross-species interactivity was tested by using *A. cellulolyticus* Xyn-DocB or (A, B) *C. thermocellum* Xyn-Doc48S (C, D). All *A. cellulolyticus* cohesins were lysate-derived, and the control cohesin, *C. thermocellum* CBM-CohA2, was affinity purified. The positive control consisted of a Xyn-CBM fusion protein, and negative control (Neg) consisted of buffer in the absence of protein.

a qualitative readout for cohesin–dockerin binding which can enable the determination of positive protein–protein interactions.

In order to test the interaction between the selected 21 cohesins and 15 dockerins, 15 cellulose-coated slides, each containing the complete cohesin library, were prepared, interacted, and probed with fluorescent antibodies. One representative fluorescence scan set (Fig. 5) shows *A. cellulolyticus* XDoc-Xyn10 interaction with the 21 *A. cellulolyticus* cohesins. This intriguing cohesin–dockerin pair involves the interaction of an enzyme that contains a type-II X-dockerin modular dyad, rather than a type-I dockerin as in most enzymes. In this case, the question arises whether the interaction with the cohesin will correspond to the normal type-II interaction scheme or whether this particular X-dockerin will interact differently. As shown in the initial data presented for the representative type-I interaction (Fig. 4), the Cy5 fluorescence channel in the more extensive set (Fig. 5A) shows the *general extent* of CBM-Coh immobilization to the cellulose slide and serves as a measure of the protein expression in the host-cell extract. The Cy3 fluorescence channel shows the extent of Xyn-XDoc immobilization on the same cellulose slide and serves as a measure of the *designated affinity-based* cohesin–dockerin interaction (Fig. 5B).

The intensity of each cohesin–dockerin interaction was normalized for protein expression by dividing each spot's Cy3 fluorescence value by its Cy5 fluorescence value.

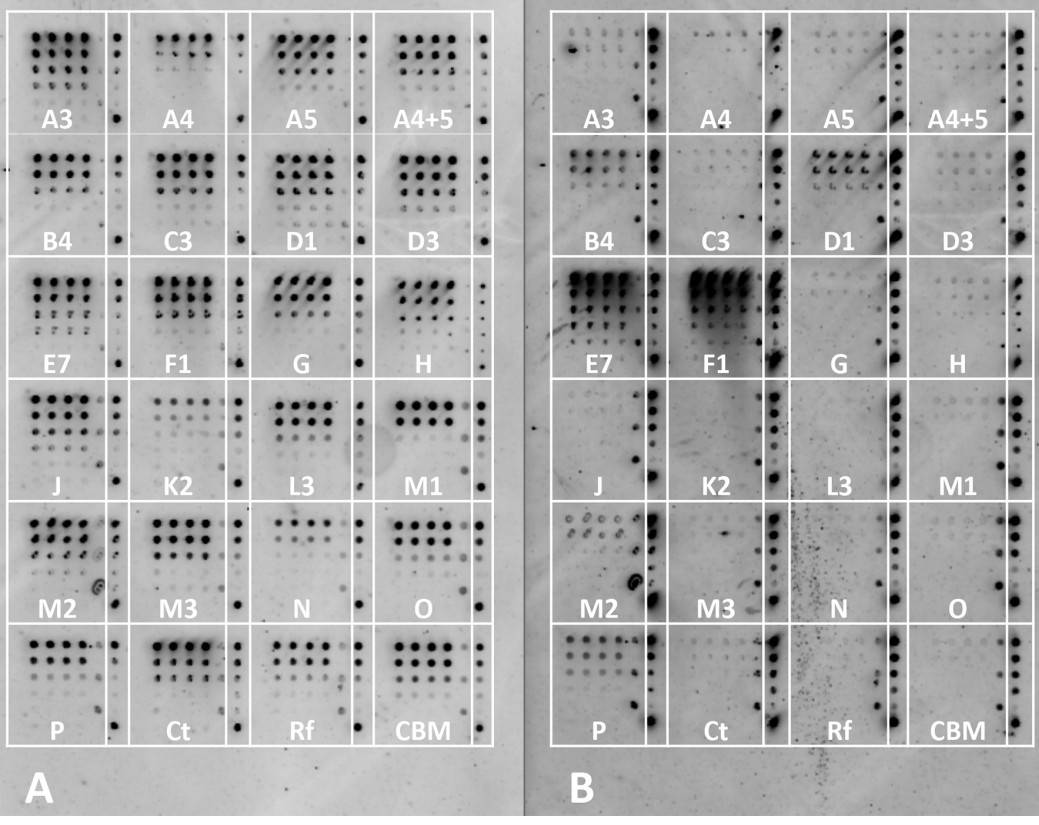

**Figure 5 Example of a comprehensive cohesin library screening, using a type-II X-dockerin dyad as a probe.** Fluorescent scans of cellulose-coated slides containing the cohesin library interacted with XDoc-Xyn10 (an X-dockerin modular dyad derived from an *A. cellulolyticus* GH10 xylanase). Scans show Cy5 anti-CBM signal (A) and Cy3 anti-xylanase signal (B). The boxes are labeled showing each *A. cellulolyticus* CBM-Coh library member (from the 3rd cohesin of scaffoldin A, designated A3 through the cohesin from scaffoldin P). Each box contains five four-fold serial dilutions of CBM-Coh, blotted vertically in quadruplicate. Two control boxes (Ct and Rf) were included on each slide to check for cross-species binding, and were labeled with CBM-Coh from *C. thermocellum* and *R. flavefaciens*, respectively. The CBM box contains only a CBM and serves as a negative control. To the right of each box, a rectangle indicating the serial dilutions of the CBM-Xyn positive control is shown. The bottom spot of the CBM-Xyn rectangle is a duplicate of the highest concentration.

Cohesin-dockerin interaction intensities were calculated for the highest two-protein dilutions, and the averaged values were graphed in a histogram. A sample histogram showing the XDoc-Xyn10 interaction with the 21 cohesin library and the control proteins is presented (Fig. 6). The interaction intensity of the highest-interacting cohesin (CohE7) is arbitrarily set to a value of 1. In this representative experiment, seven cohesins (CohA4, CohB4, CohD1, CohD3, CohF1, CohM1 and CohP) were found to interact to varying extents with XDocXyn10. The averaged cohesin–dockerin interaction intensities for all the interactions were completed by repeating this process with all 15 dockerins.

The complete data sets were then normalized and compiled (Table 1).

Previously determined specificities of cohesin–dockerin interactions were generally confirmed in this work. The type-I interactions between CohA3, CohA4, CohA5 (and the

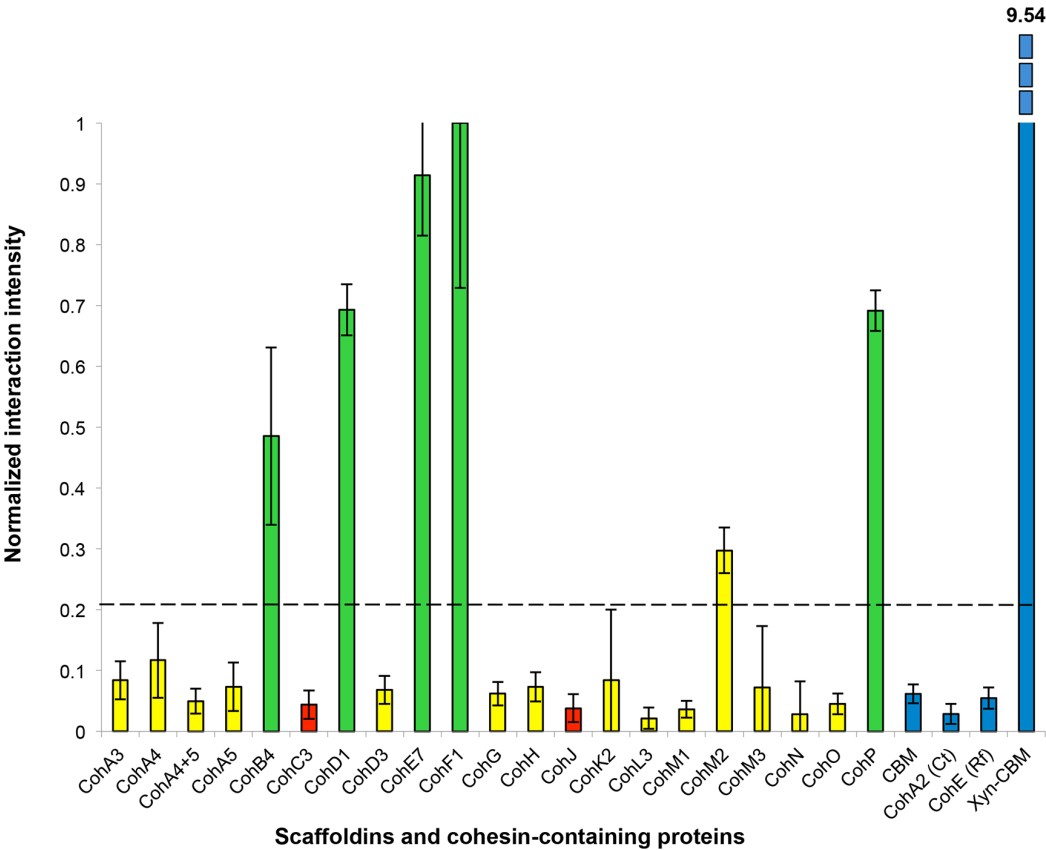

**Figure 6 Representative histogram showing cohesin library screen results.** Normalized interaction intensities between the *A. cellulolyticus* CBM-Coh library and XDoc-Xyn10. CBM-Cohs A3 though P are shown along with negative controls (CBM alone, *C. thermocellum* CBM-CohA2, *Ruminococcus flavefaciens* CBM-CohE), and the positive control (Xyn-CBM). CohF1 was found to have the highest experimental interaction intensity and was normalized to an interaction value of one. The dashed line indicates the threshold for positive interaction, determined by the CBM negative control.

CohA4-CohA5 modular dyad), as well as CohD3, all showed defined binding specificity to the type-I dockerins of Cel48A, Cel124A and ZP_09465673. Additional type-I cohesins, CohG, CohH, CohM1 and CohM3 showed a similar binding pattern as revealed in this work, although CohH appeared to be more selective in binding to the dockerin of Cel48A.

As anticipated (*Xu et al., 2003*), the type-I CohC3 bound exclusively to the dockerin of ScaB. The same specificity pattern was also demonstrated for CohJ. Surprisingly, the CohJ sequence shows close similarity to CohH and other type-I cohesins, such as the ScaA cohesins, but is relatively distant to those of the ScaC cohesins (*Dassa et al., 2012*).

The type-II cohesins examined in this work tended to bind selectively to the type-II X-dockerins of ScaA and Xyn10 (ZP_09461840), although CohB4 and CohP bound only to the latter.

Several of the predicted cohesins and dockerins failed to bind in a clearly selective manner to any of the modular partners tested in this work (Table 1). These include the type-I cohesins CohK2, CohL3, CohN and CohO, the scaffoldin-borne dockerins DocK1,
Table 1 **Summary of *A. cellulolyticus* cohesin–dockerin interactions.** Interaction matrix showing the binding intensities of the complete cohesin–dockerin library with cohesins and dockerins presented by rows and columns, respectively. The protein–protein interactions are color coded based on their measured interaction above the background cutoff threshold as indicated in the gray scale key, for example 3x denotes an interaction 3-times over the threshold interaction level. The positive control interaction is between a *C. thermocellum* cohesin and dockerin (from the Cel48 cellobiohydrolase).

| | DocCel48 ZP_09463651 | DocCel124 ZP_09464332 | Doc ZP_09465673 | XDocA | XDoc-Xyn10 ZP_09461840 | DocB | DocK1 | DocK2 | DocO | XDocP | DocCel9 | Doc-CBM ZP_09463099 | Doc-domain ZP_09463877 | Doc-peptidase ZP_09464147 | Doc_Doc ZP_09465996 | Positive Control |
|---|---|---|---|---|---|---|---|---|---|---|---|---|---|---|---|---|
| CohA3 | 3x | 3x | 2x | | | | | | | | | | | | | |
| CohA4 | 5x | 3x | 3x | | | | 1x | | | | | | | 1x | | |
| CohA5 | 3x | 3x | 3x | | | | | | | | | | | | | |
| CohA4+5 | 4x | 3x | 3x | | | | | | | | | | | | | |
| CohD3 | 3x | 3x | 2x | | | | | | | | | | | | | |
| CohG | 2x | 2x | 2x | | | | | | | | | | | | | |
| CohH | 2x | | 1x | | | | | | | | | | | 1x | | |
| CohM1 | 3x | 2x | 2x | | | | | | | | | | | | | |
| CohM2 | 4x | 3x | 2x | 2x | 1x | 1x | 1x | 1x | 3x | 3x | | 3x | 3x | 3x | 1x | 1x |
| CohM3 | 2x | 2x | 2x | | | | | | | | | | | | | |
| CohB4 | | | 1x | | 2x | | | | | | | | | | | |
| CohD1 | 1x | | 2x | 3x | 3x | 1x | | 1x | 2x | 2x | 1x | 2x | 1x | 2x | | |
| CohE7 | | | | 3x | 4x | | | | | | | | | | | |
| CohF1 | | | 2x | 3x | 4x | | | 1x | 1x | | | | | 1x | | |
| CohP | | | | 1x | 3x | | | | | | | | | | | |
| CohC3 | | | | | 5x | | | | | | | | | | | |
| CohJ | | | | | 3x | | | | | | | | | | | |
| CohK2 | | | | | | | | | | | | | | | | |
| CohL3 | 1x | | | | | | | | | | | | | | | |
| CohN | | | | 1x | | | | | | 1x | | | | | | |
| CohO | | | | | | | | | | | | | | | | |
| Control | | | 1x | | | | | | | | | | | | | 5x |

| Affinity profile 1 | Affinity profile 2 | Affinity profile 3 |
|---|---|---|

| Measured interaction intensity above background cutoff threshold | | | | |
|---|---|---|---|---|
| ~1x cutoff | ~2x cutoff | ~3x cutoff | ~4x cutoff | ~5x cutoff |

DocK2, DocO, the type-II XDocP as well as the type-I Cel9 dockerin and those of the following proteins: ZP_09463099, ZP_09463877, ZP_09464147 and ZP_09465996. In addition, cohesins CohM2, CohD1 and, to a lesser extent, CohF1 showed significantly high levels of nonspecific binding to numerous dockerins. No clear selectivity pattern could be assigned to CohM2, despite its somewhat higher binding to the Cel48S dockerin and clear sequence similarity to its neighboring ScaM cohesins (CohM1 and CohM3), which indeed displayed clear binding specificity for selected type-I dockerins. Despite the observed nonspecific binding pattern, CohD1 and CohF1 showed somewhat greater levels of binding compared to the aforementioned type-II X-dockerins.

## CONCLUSIONS

The cellulosomes of anaerobic cellulolytic bacteria comprise an incredibly intricate set of components which can self-assemble into a variety of mature complexes. In this study, we developed a standardized cloning, expression, testing, and analysis procedure in order to determine how a representative set of cohesin–dockerin interactions in a given bacterium can be arranged into complex cellulosomal architectures. For this purpose, we used the newly sequenced genome of *A. cellulolyticus*—a bacterium known to produce an exceptionally elaborate cellulosome system—as an example of how one can approach this issue. Through this "quick and dirty" system, we identified 90 positive interactions from the 315 representative interactions tested (Table 1). The great majority of the positive interactions conformed to the predicted affinity types between cohesins and dockerins, based on their amino acid sequences. However, a few cohesins appear to exhibit non-specific binding under our experimental conditions (see below). For the cohesins and dockerins displaying the expected type-specific interactions, three affinity profiles were apparent. By compiling the connectivity data for cohesin–dockerin pairs that were shown to be moderately to highly interactive, expanded models of several different cellulosome assemblies were constructed, representing both cell-bound cellulosome systems (Fig. 7A) and cell-free cellulosome systems (Fig. 7B). The connectivities between cellulosome components are shown by color-coded protein modules, brackets, and arrows, in correspondence with the affinity profile.

**Affinity profile 1**, shown in yellow (Figs. 7A and 7B), consists of three dockerins (DocCel48, DocCel124, and ZP_09465673) and their binding partners. These dockerins are found on CBM- and enzyme-containing proteins and generally recognize type-I cohesins. These larger cellulosome systems tend to comprise the periphery of the cell-bound cellulosomes, but the individual type-I dockerin-bearing proteins can also attach directly to the cell-bound scaffoldins, such as ScaD and ScaG. In the cell-free cellulosome system, affinity profile-1 dockerins were confirmed to bind the cohesins on the primary scaffoldin ScaM, which bears a pair of cellulose-binding family 2 CBMs.

**Affinity profile 2**, shown in green (Figs. 7A and 7B), consists of two related X-dockerin modular dyads examined in this work (XDocA and XDocXyn10) and their binding partners. XDocXyn10 showed strong interactions with cohesins that were designated as type II, based on amino acid sequence (cohesins B4, D1, E7, F1, and P). However,

## A. Cell-bound cellulosome systems

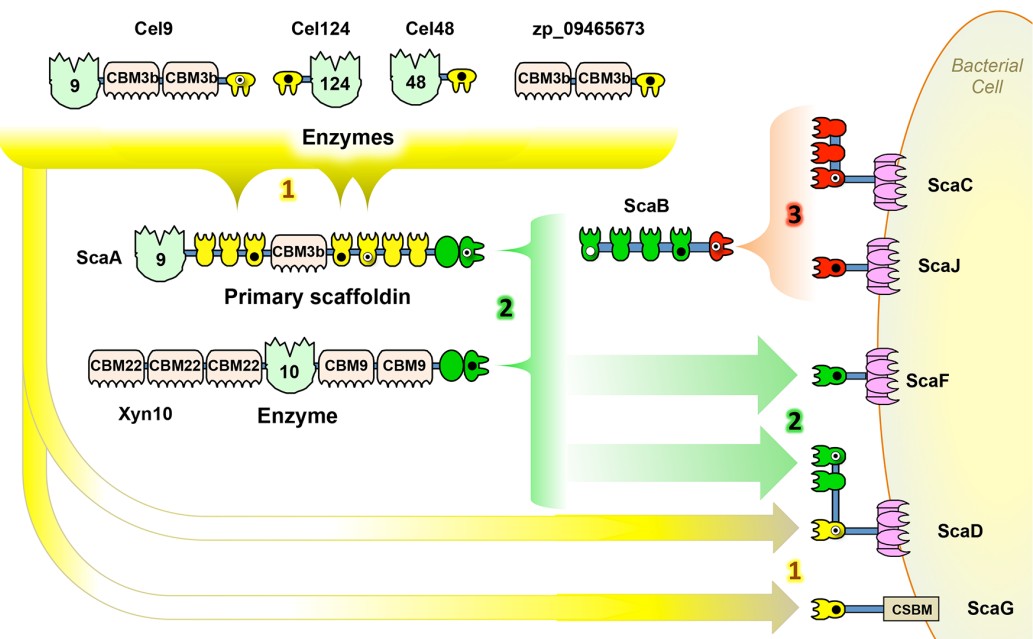

## B. Cell-free cellulosome systems

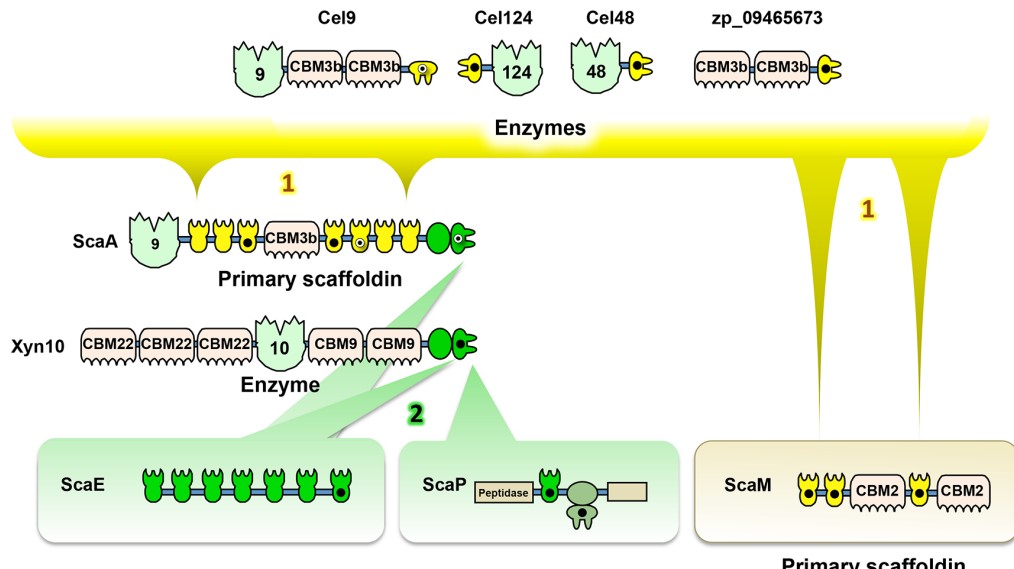

**Figure 7 Architectural models of *A. cellulolyticus* cellulosome systems. (A) Cell-bound cellulosome systems. (B) Cell-free cellulosome systems.** Confirmed cohesin–dockerin interactions (those interactions ∼2x above the background cutoff threshold) are indicated in the connectivity scheme. Color coding (yellow, green and red) of the modules, brackets, and arrows reflect the respective affinity profiles enumerated in the figure. Black circles, white circles, and black/white circles indicate modules studied in this work, in previous works, and in both this work and previous works, respectively.

XDocA only showed strong interactions with cohesins D1 and E7 and F1. In the cell-bound cellulosome systems, affinity profile 2 cohesins and dockerins can be found either on peripheral cellulosome components (such as Xyn10) or on intermediate components that connect peripheral to surface-bound components (such as ScaA and ScaB), as well as on cell–surface bound components themselves (such as ScaF and ScaD). In the cell-free cellulosome system, affinity profile 2 components were confirmed to strongly interact with the heptavalent scaffoldin, ScaE, as well as with the unusual peptidase-containing ScaP. The previously presumed binding (*Xu et al., 2003*) between XDocA and CohB4 was not observed in this study.

**Affinity profile 3**, shown in red (Fig. 7A), consists solely of DocB and its binding partners on the surface-anchored scaffoldins ScaC and ScaJ. Our screen confirmed the role of affinity profile 3 components' role as adaptors between the cell surface and intermediate scaffoldin ScaB. The binding between the ScaB dockerin and ScaC cohesins was reported earlier (*Xu et al., 2003*); the screening procedure documented in this work successfully revealed a similar binding specificity between the ScaB dockerin and the single ScaJ.

## Cellulosome architectures

This study was intended to give a preliminary view of the 5904 possible interactions of the *A. cellulolyticus* cellulosome system and covered approximately 7% of all theoretical cohesin–dockerin interactions. Although the study's selective screening approach allowed for a diverse and representative library to be constructed, many important interactions of the *A. cellulolyticus* cellulosome undoubtedly remain to be discovered. Notwithstanding this, the study greatly expanded our knowledge of the *A. cellulolyticus* cellulosomal architectures.

This study revealed that in addition to the previously demonstrated cellulosomes A and B (Fig. 1), based on ScaC and ScaD, respectively, several additional cell-bound cellulosome systems exist in *A. cellulolyticus*. While ScaC contains three cohesins, and can form large cellulosomes by utilizing the adaptor scaffoldin ScaB, the novel scaffoldin ScaJ contains only one cohesin and can form a smaller version of the ScaB-containing cellulosome. The bacterium's ability to modulate from a large cellulosome based on ScaC to a similar, but smaller one, based on ScaJ may give some insight into how cellulosome size and architecture can be adjusted based on the bacterium's environmental requirements.

The previously reported ScaD contains both type-I and type-II cohesin functionalities, allowing for the construction of complex cellulosome systems (*Xu et al., 2004*). The present study confirmed that the novel ScaF and ScaG can facilitate the assembly of cellulosomes that are similar, but contain only one of the affinity types (Fig. 7A). In comparison to cellulosomes based on ScaD, these novel cellulosomes are simpler. Utilizing type-II connectivity, ScaF can facilitate a connection to either Xyn10 or to the primary scaffoldin ScaA, which in turn is capable of arraying a variety of enzymes. ScaG can form the simplest cell-bound system by directly binding individual enzymes through its single type-I cohesin. Interestingly, ScaG utilizes a novel surface-binding module similar to that of the OlpC scaffoldin of *C. thermocellum* (*Pinheiro et al., 2009*). These cellulosome models

demonstrate that *A. cellulolyticus* is capable of synthesizing cellulosomes in a range of sizes and complexities to suit its particular needs.

This study also tested modules in *A. cellulolyticus*'s cell-free cellulose systems. The base scaffoldin ScaE can form soluble cellulosomes based on a type-II interaction, allowing the attachment of multiple Xyn10 and ScaA scaffoldins, which in turn can bind a complement of 7 type-I dockerin-containing enzymes. These cellulosomes are similar to those enabled by ScaF, although they are larger and cell-free. Because ScaE is capable of binding these enzymes, it allows the bacterium to project its complement of enzymatic activity far from the cell surface.

The only binding partner identified so far for ScaP's cell-free cellulosome system is Xyn10, which binds to ScaP through the type-II cohesin. The unusual XDoc adjacent to this cohesin was not found to bind with any cohesin selected for this screen. The presence of a peptidase (an enzyme not normally found in cellulosomes) and an additional unknown domain highlight the need for further study to understand the many functionalities present in many of the components of the cellulosome.

A cell-free primary scaffoldin ScaM was also found to form complexes with multiple type-I dockerins through all three of its cohesins. This scaffoldin represents a larger, cell-free relative to the cell-bound ScaG. Taken together, the complement of cell-bound and cell-free cellulosome systems revealed in this study demonstrates that *A. cellulolyticus* has the ability to synthesize a broad range cellulosomes that complement one another in size, complexity, and activity on or away from the cell.

## Comparison to previous interaction studies

Previous work on the adaptor scaffoldin ScaB (*Xu et al., 2003*) showed strong binding between its N-terminal cohesin (CohB1) and the dockerin of ScaA, XDocA. At the time, XDocA was assumed to bind all four cohesins of ScaB. To test this we included CohB4, the fourth cohesin of ScaB, in our screen. We found only weak binding between CohB4 and XDocA, demonstrating that the cohesins of ScaB may possess differing binding specificities. Future studies should thus consider the binding specificities of all four ScaB cohesins. The interaction patterns between XDocA and DocB and other previously studied cohesins (CohA5, CohC3, CohD1, and CohD3) were also confirmed by our method.

Although in general, our results conformed with our predictions about cohesin–dockerin specificity, some anomalous results pointed to some tradeoffs in our "quick and dirty" method. One potential challenge associated with our experimental approach is the use of recombinant proteins and the *E. coli* host. Although precautions were taken in order to clone the desired module into the expression cassette, it is possible that some of the expressed modules did not include the complete sequence or that they require additional protein components in order to function correctly. Additionally, *E. coli* may not be completely equipped to properly express some of the *A. cellulolyticus* proteins. As stated earlier, many of the proteins tested did not show substantial binding to any of the selected binding partners. It is possible that this resulted from a genuine lack of binding partner, or because the *E. coli* host was unable to properly express one or both of the

binding partners in an active folded state. Nonetheless, this experimental approach has revealed new and interesting cohesin–dockerin binding partners.

Additionally, of the 21 cohesins tested, CohD1 and CohM2 showed significant nonspecific binding profiles. It is possible that these proteins are indeed promiscuous binders, but it is also possible that due to the expression or experimental conditions, the binding specificities of these modules have been compromised. Consequently, inclusion of these cohesins in the cellulosome models should be viewed with a degree of uncertainty. The nonspecific binding of CohM2 was surprising, given its sequence homology to its neighboring cohesins, CohM1 and CohM3.

Based on previous studies, our method produced at least one false-negative interaction result. Cel9 was previously shown to interact with CohA5 and CohD3, but our experiments did not replicate these results. Since our system does not optimize protein expression conditions, it is possible that some proteins are not properly folded and are unsuitable for binding or that some are not present at appropriate protein-binding concentrations. Future iterations of this method should vary protein expression conditions in order to allow more of the proteins to express correctly and to be tested.

Although the approach described in this work is prone to some inconsistencies, which include false negatives and false positives, it can provide a preliminary overview of the profile of cohesin–dockerin interconnectivities, and consequently of the cellulosomal architecture of a given species. It is clear that further experiments would be required in order to reliably confirm these interactions.

Through the results of this selective screen of cohesin–dockerin binding interactions of *A. cellulolyticus*, we have begun to better understand the details of its elaborate surface-bound cellulosomes (*Lamed et al., 1987*) and can now begin to understand how its complement of enzymes synergize to accomplish effective cellulolytic activity. Before the sequencing of the *A. cellulolyticus* genome, its cellulosome systems, although considered elaborate compared to others, were only partially understood, and comprised only two cellulosome architectures. With the sequencing of the complete genome and subsequent experimental analysis, 12 additional putative scaffoldins were discovered, as well as many other cellulosomal components (*Dassa et al., 2012*). The results of the current work have begun to explain how these many components interconnect to form one of the most complex cellulosome systems yet discovered. The approach featured in this work will serve as a platform to more fully understand the architectural diversity in the ever-growing field of cellulosomics.

### Funding

This research was supported by a grant (No. 1349) to EAB from the Israel Science Foundation (ISF) and a grant (No. 24/11) issued to RL by The Sidney E. Frank Foundation through the ISF. Additional support was obtained from the Israeli Center of Research Excellence (I-CORE Center No. 152/11) managed by the Israel Science Foundation, from

the United States-Israel Binational Science Foundation (BSF), Jerusalem, Israel, by the Weizmann Institute of Science Alternative Energy Research Initiative (AERI) and the Helmsley Foundation. The authors also received support from the European Union, Area NMP.2013.1.1-2: Self-assembly of naturally occurring nanosystems: CellulosomePlus Project number: 604530 and an ERA-IB Consortium (EIB.12.022), acronym FiberFuel. In addition, EAB received a grant from the F. Warren Hellman Grant for Alternative Energy Research in Israel in support of alternative energy research in Israel administered by the Israel Strategic Alternative Energy Foundation (I-SAEF). E.A.B. is the incumbent of The Maynard I. and Elaine Wishner Chair of Bio-organic Chemistry. The funders had no role in study design, data collection and analysis, decision to publish, or preparation of the manuscript.

### Grant Disclosures

The following grant information was disclosed by the authors:
Israel Science Foundation (ISF): 1349.
The Sidney E. Frank Foundation: 24/11.
Israeli Center of Research Excellence.
United States-Israel Binational Science Foundation (BSF).
Weizmann Institute of Science Alternative Energy Research Initiative (AERI).
The Helmsley Foundation.
The European Union: 604530.
The F. Warren Hellman Grant for Alternative Energy Research in Israel.
Israel Strategic Alternative Energy Foundation (I-SAEF).

### Competing Interests

Edward A. Bayer is an Academic Editor for PeerJ.

### Author Contributions

- Yuval Hamberg conceived and designed the experiments, performed the experiments, analyzed the data, contributed reagents/materials/analysis tools, wrote the paper, prepared figures and/or tables, reviewed drafts of the paper.
- Vered Ruimy-Israeli conceived and designed the experiments, performed the experiments, contributed reagents/materials/analysis tools, wrote the paper.
- Bareket Dassa and Edward A. Bayer conceived and designed the experiments, wrote the paper, prepared figures and/or tables, reviewed drafts of the paper.
- Yoav Barak conceived and designed the experiments, contributed reagents/materials/analysis tools, wrote the paper.
- Raphael Lamed wrote the paper.
- Kate Cameron and Carlos M.G.A. Fontes wrote the paper, reviewed drafts of the paper.
- Daniel B. Fried wrote the paper, prepared figures and/or tables, reviewed drafts of the paper.

## Data Deposition

The following information was supplied regarding the deposition of related data:

https://drive.google.com/file/d/0By1mO_hlvksic0Y1TDcwQzdON3c/edit?usp=sharing

## Supplemental Information

Supplemental information for this article can be found online at http://dx.doi.org/10.7717/peerj.636#supplemental-information.

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
