# Peer review of "Elaborate cellulosome architecture of Acetivibrio cellulolyticus revealed by selective screening of cohesin–dockerin interactions"

_PeerJ, doi:10.7717/peerj.636_

## Round 0.1 · original submission · Minor Revisions

Dear Ed,

Your paper has been reviewed by four experts in the field, all found your paper interesting and recommended to accept the paper after some minor issues are addressed.

Therefore, I invite you to revise the paper based on these minor requests. I look forward to receiving your revised paper very soon.

best regards

George

George Guo-Qiang CHEN (Ph.D.)
Professor of Microbiology and Biomaterials
Department of Biological Sciences and Biotechnology
School of Life Sciences
Tsinghua University
Beijing 100084 China
Tel: +86-10-62783844
Fax: +86-10-62794217
e-mail: [email protected]

·

Basic reporting

This work proposed the strategy to screen cohesin-dockerin interaction and validate the information obtained by bioinformatic tool, which can lead to high-throughput screening of the microbial whole genome. Much effort was made to elucidate the role and function of dockerin-cohesin interaction, which finally lead us to get the picture of cellulosome assembly—interlocking several building blocks like “Lego” construction. This work provides better understanding of and extends our knowledge on cellulosome architectures.

Experimental design

Experiments were carefully designed and methodology can be used as a guideline to study diversity of cohesin-dockerin interaction from different microbes.

Validity of the findings

Importance in its field (biomass conversion/biocatalysts) and extend knowledge on beautifully natural-occurring cellulosmes evolved by nature.

Additional comments

The results gained from this work are significant to cellulosome paradigm, which is related to plant cell wall degradation strategies—important for sustainable biofuel/chemical production from lignocellulose. However, I have found some erroneous in figure number running (sequence) that need minor modification.

The sequence of figure number mentioned here is based on those in the manuscript sent to review.

Line 220: Figure 4 doesn’t show the dockerin sequences as you mentioned in the text. Is Figure 7 you refer to? By the way Figure 7 is similar to Dassa et al., BMC Genomics 2012, 13:210. Did you modify from their work?
Line 259: (Figures 5A and 5B) should likely be Figures 4A and 4B; the explanation matches to them. By the way there is some intensity on 4B Ct A2. Does it mean the purified A. cellulolyticus DocB could bind to C. thermocellum CohA2?
Line 262: (Figures 5C and 5D) should likely be Figures 4C and 4D?
Line 288: Figure 6 should be Figure 5?
Line 294: Figure 5 should be Figure 4 and Figure 6A should be Figure 5A?
Line 298: Figure 6B should be Figure 5B?
Line 303-304: Figure 7 should be Figure 6?

Figure 4
Line 617-618: A. cellulolyticus Xyn-DocB or (C, D) C. thermocellum Xyn-Doc48S (A, B) should be A. cellulolyticus Xyn-DocB (A, B) or C. thermocellum Xyn-Doc48S (C, D)

Reviewer 2 ·

Basic reporting

The manuscript by Hamberg and coworkers describes the assessment of cohesin-dockerin interactions in Acetivibrio cellulolyticus.
This report includes interesting data, and as mentioned in the paper, the authors revealed not only several cell bound cellulosome architecture but also soluble cellulose systems that are not bound to the bacterial cell surface.

Experimental design

This paper presents a well-conducted, carefully controlled set of assays. The focus of the work is clear right from the beginning, and the data fully support the final conclusions. Data are well presented.

Validity of the findings

Page 6, line 132, “BamHI and XhoI” “HI” and “I” should not be italic font.
Page 6, line 135, Please specify the concentration of kanamycin.

Reviewer 3 ·

Basic reporting

No comments with respect to adherence to PeerJ policies.

Experimental design

The novelty in the presented experimental design and approach is the strongest point of the manuscript that fully justifies its publication (use of whole-cell extracts and affinity interactions). The relevant procedures are clearly described and the method is applied at maximum reliability including all appropriate controls.

Validity of the findings

The obtained results are presented as clearly as possible in the relevant figures and tables despite the complexity and the number of the interactions studied (21 cohesins x 15 dockerins). The proposed methodology proved quite reliable for the majority of the studied interactions while the small number of pitfalls observed are clearly discussed. In addition it was able to reveal several new interaction among the studies cellulosomal modules. This is quite important taking into account the great complexity of the A. cellulolyticus cellulosome system.

Reviewer 4 ·

Basic reporting

Using the described newly-designed screening methods, some novel architectures were found in cell-bound cellulosomes of Acetivibrio cellulolyticus, as well as in some cell-free cellulosome systems produced by this organism. These findings not only expand our knowledge in the field of cellulosome structure, but also could help to design new artificial cellulosomes to convert lignocellulose to biofuels efficiently. This manuscript is written in good English; all descriptions including background introduction in text was clear.

Experimental design

Using whole cell-lysate, instead of purifying individual proteins, to screen cohesin-dockerin interactions in large numbers is a new and time-saving approach introduced in this study. Experiments were designed well, data collection was appropriate and sufficient to support the hypothesis, and the description of the methods is clear and detailed.

Validity of the findings

The collected data is sufficient to support the conclusions, and the conclusions are stated carefully and appropriately. Some of the conclusions derived from this study are new and expand our knowledge not only for cellulosomes of the bacterium Acetivibrio cellulolyticus, but also for the whole field of cellulosome chemistry. This study will therefore have important effects on the further study of cellulosomes in theory and application in the future.

Additional comments

Some suggestion for minor revision:
1. Line 50, “…as well as soluble cellulose systems that are not…” “cellulose” should be changed to “cellulosome”.
2. Line 84, “…dockerin of ScaA (Ding et al. 1999).” “(Ding et al. 1999)” should be changed to “Xu et al. 2003”.
3. Line 88, “a cellulose-binding module (CBM) (Boraston et al. 2004).” Please double check this cited reference to ensure you really want to cite it here.
4. Line 178, “, with the highest measurement…” which one is the highest measurement? Is the fusion-protein positive control included in the competition for the highest ratio?
5. Line 191-192, “From the total 5904 possible cohesin dockerins of A. cellulolyticus…” could be changed to “From the total 5904 possible cohesin-dockerin combinations presented by A. cellulolyticus,” Also, how to calculate to get 5904? 143 dockerins times 41 cohesins implies only 5863 combinations.
6. Line 262, “…(Figure 5C and 5D).” These two figures cannot be found in the PDF file.
7. Line 330, if “although” were changed to “despite”, this sentence would state more clearly what I assume it means.
8. Line 196-197, this sentence could be changed to “The scaffoldins and other cohesin-containing proteins of A. cellulolyticus, along with nine additional…
9. Line 360-361, “but can also … scaG.” this sentence could be changed to “but the individual type-I-dockerin-bearing proteins can also attach directly to the cell-bound scaffoldins, such as ScaD and scaG.” Remove “Additionally, ScaD, a surface-anchored scaffoldin also binds these components.”, as it is repetitious.
10. Line 372, according to Figure 8, “ScaP” is in the cell-free cellulosome system, NOT in the cell-bound cellulosome systems.
11. Line 375, “…scaffoldin, ScaE” could be changed to “…scaffoldin, ScaE as well as with the unusual peptidase-containing ScaP”.
12. Lines 576-577 (Figure 1), in the legend for this figure, the doubled red-yellow icons shown for type I cohesins and dockerins do not appear anywhere in the figure itself. Instead, the type I modules are shown in the figure as yellow icons.
13. Line 584, Why is ScaC called a “type I scaffoldin”, when it has all type-II cohesins?
14. Line 636, “between between”, please remove one “between”.

---

## Round 0.2 · accepted · Accept

Dear Ed,

I am pleased to inform you that your revised paper is now accepted by PeerJ.

Regards

George


George Guo-Qiang CHEN (Ph.D.)
Professor of Microbiology and Biomaterials
PeerJ Editor
Department of Biological Sciences and Biotechnology
School of Life Sciences
Tsinghua University
Beijing 100084 China
Tel: +86-10-62783844
Fax: +86-10-62794217
e-mail: [email protected]